# Restoration of *Mecp2* expression in GABAergic neurons is sufficient to rescue multiple disease features in a mouse model of Rett syndrome

Kerstin Ure[1,2], Hui Lu[1,2,3], Wei Wang[1,2], Aya Ito-Ishida[1,2], Zhenyu Wu[2,4], Ling-jie He[1,2,3], Yehezkel Sztainberg[1,2], Wu Chen[2,5,6], Jianrong Tang[2,4], Huda Y Zoghbi[1,2,3,5]*

[1]Department of Molecular and Human Genetics, Baylor College of Medicine, Houston, United States; [2]Jan and Dan Duncan Neurological Research Institute, Texas Children's Hospital, Houston, United States; [3]Howard Hughes Medical Institute, Baylor College of Medicine, Houston, United States; [4]Department of Pediatrics, Baylor College of Medicine, Houston, United States; [5]Department of Neuroscience, Baylor College of Medicine, Houston, United States; [6]Cain Foundation Laboratories, Baylor College of Medicine, Houston, United States

**Abstract** The postnatal neurodevelopmental disorder Rett syndrome, caused by mutations in *MECP2*, produces a diverse array of symptoms, including loss of language, motor, and social skills and the development of hand stereotypies, anxiety, tremor, ataxia, respiratory dysrhythmias, and seizures. Surprisingly, despite the diversity of these features, we have found that deleting *Mecp2* only from GABAergic inhibitory neurons in mice replicates most of this phenotype. Here we show that genetically restoring *Mecp2* expression only in GABAergic neurons of male *Mecp2* null mice enhanced inhibitory signaling, extended lifespan, and rescued ataxia, apraxia, and social abnormalities but did not rescue tremor or anxiety. Female *Mecp2*[+/-] mice showed a less dramatic but still substantial rescue. These findings highlight the critical regulatory role of GABAergic neurons in certain behaviors and suggest that modulating the excitatory/inhibitory balance through GABAergic neurons could prove a viable therapeutic option in Rett syndrome.

*For correspondence: hzoghbi@bcm.edu

**eLife digest** Rett syndrome is a childhood brain disorder that mainly affects girls and causes symptoms including anxiety, tremors, uncoordinated movements and breathing difficulties. Rett syndrome is caused by mutations in a gene called *MECP2*, which is found on the X chromosome. Males with *MECP2* mutations are rare but have more severe symptoms and die young. Many researchers who study Rett syndrome use mice as a model of the disorder. In particular, male mice with the mouse equivalent of the human *MECP2* gene switched off in every cell in the body (also known as *Mecp2*-null mice) show many of the features of Rett syndrome and die at a young age.

The *MECP2* gene is important for healthy brain activity. The brain contains two major types of neurons: excitatory neurons, which encourage other neurons to be active; and inhibitory neurons, which stop or dampen the activity of other neurons. In 2010, researchers reported that mice lacking *Mecp2* in only their inhibitory neurons develop most of the same problems as those mice with no *Mecp2* at all. This discovery led Ure et al. – including a researcher involved in the 2010 study – to ask if activating *Mecp2* in the same neurons in otherwise *Mecp2*-null mice was enough to prevent some of their Rett syndrome-like symptoms.

The experiments showed that male mice that only have *Mecp2* activated in their inhibitory neurons lived several months longer than male *Mecp2*-null mice. These male "rescue mice" also moved normally and had a normal body weight, though they still experienced anxiety, tremors and breathing difficulties. Female mice represent a better model of human Rett syndrome patients, and Ure et al. found that female rescue mice showed smaller improvements than the males.

These data suggest that when a brain is missing *Mecp2* everywhere, as in male *Mecp2*-null mice, turning on *Mecp2* in inhibitory neurons can make the brain network nearly normal and prevent most Rett-syndrome-like symptoms. However, the brains of female rescue mice contain both normal cells and cells with mutated *Mecp2*. This mixture of normal and abnormal cells appears to cause abnormalities that cannot be overcome by rescuing just the activity of the inhibitory neurons. These findings also highlight the importance of doing future studies in female mice to better understand the development of Rett syndrome.

The next challenge is to test different ways of activating the inhibitory neurons in the female mouse brain, for example by using drugs that target these neurons. It is hoped these methods will help researchers to refine a path toward potential new treatments for Rett syndrome patients. Finally, in a related study, Meng et al. asked how deleting or activating *Mecp2* only in the excitatory neurons of mice affected Rett-syndrome-like symptoms.

## Introduction

Maintaining a proper ratio of excitation and inhibition throughout the brain is critical to normal neurological function. Alterations in this excitatory/inhibitory balance are postulated to underlie a number of neuropsychiatric disorders, such as autism (*Vattikuti and Chow, 2010*; *Gogolla et al., 2009*; *Rubenstein and Merzenich, 2003*), schizophrenia (*Belforte et al., 2010*), and Rett syndrome (*Dani et al., 2005*). Much of the work on this balance has focused on excitatory neurons, which make up the majority of the brain's neuronal population, yet it has become increasingly clear that inhibitory neurons, which predominantly release the inhibitory neurotransmitter gamma-aminobutyric acid (GABA), play important roles in the proper function of entire circuits (*Xue et al., 2014*) and in the behaviors regulated by these circuits (*Yizhar et al., 2011*). Dysfunctional GABAergic signaling has been implicated in multiple neurological and neuropsychiatric disorders (*Siniatchkin and Koepp, 2009*; *Pizzarelli and Cherubini, 2011*), including Rett syndrome (*Chao et al., 2010*). In fact, deletion of *Mecp2* solely in GABAergic neurons is sufficient to reproduce the majority of the Rett-like features of the constitutive *Mecp2*-null mouse, including ataxia, stereotyped behaviors, seizures, breathing abnormalities, and premature death (*Chao et al., 2010*).

Rett syndrome (RTT) is caused by mutations in the X-linked gene encoding methyl CpG-binding protein 2 (MeCP2) (*Amir et al., 1999*), a protein that is highly expressed throughout the brain and involved in chromatin modulation (*Baker et al., 2013*). RTT is notable for the variety of different behaviors it affects: disease-causing *MECP2* mutations cause a rapidly fatal neonatal encephalopathy

in human males but allow females to develop normally until around one year of age, at which point they regress developmentally, losing acquired language, social, and motor milestones, and instead develop ataxia, seizures, hand stereotypies, learning and memory deficits, and respiratory abnormalities (*Trevathan and Naidu, 1988*). *Mecp2* knockout mice replicate all these features of the human disease (*Chen et al., 2001*; *Guy et al., 2001*), with male *Mecp2*-null mice developing symptoms and dying within the first five months of life. *Mecp2*-heterozygous females develop progressive neurological dysfunction with a later onset, because random X chromosome inactivation preserves expression of wildtype *Mecp2* in more than half of their cells (*Samaco et al., 2013*; *Young and Zoghbi, 2004*).

While *Mecp2*-mutant male and female mice develop profound disabilities, RTT is not a neurodegenerative disorder. *Mecp2*-null neurons are rendered partly dysfunctional but are still present, active, and properly located developmentally (*Chao et al., 2007*; *2010*); this makes the RTT mouse model ideal for interrogating how certain cell types affect behavior by bypassing the confound of widespread cell death and modeling non-neurodegenerative neuropsychiatric disorders more accurately. This system has already been used to demonstrate the critical role for GABAergic signaling in the pathogenesis of RTT (*Chao et al., 2010*), as well as revealing non-overlapping roles for parvalbumin (PV)- and somatostatin (SOM)-expressing GABAergic neurons in behavior (*Ito-Ishida et al., 2015*), a role in feeding and aggression for *Sim1*-positive hypothalamic neurons (*Fyffe et al., 2008*), and a critical role for proper glutamatergic function in anxiety, tremor, and acoustic startle response (see companion paper *Meng et al., 2016*). Conversely, rescue of *Mecp2* expression in certain cell types can indicate whether targeting that cell type has therapeutic potential and can give insight into the interplay between multiple circuits in the diseased brain. Therefore, we genetically re-expressed *Mecp2* solely in GABAergic neurons in a mouse otherwise null for *Mecp2*, restoring *Mecp2* expression and function to inhibitory neurons, and observed profound improvement in both male and female mice.

## Results

### Male GABAergic rescue mice have normalized body weight and extended lifespan

Our lab previously generated a mouse line in which Cre expression is driven by the *Slc32a1* promoter (*Slc32a1*-Cre, referred to as *Viaat*-Cre henceforth), limiting Cre expression to GABAergic and glycinergic neurons (*Chao et al., 2010*). To confirm that *Viaat*-Cre was not expressed in glial cells or microglia (*Lioy et al., 2011*; *Derecki et al., 2012*; *Wang et al., 2015*), we crossed *Viaat*-Cre male mice with female ROSA-LSL-YFP transgenics and co-stained for glial fibrillary acidic protein (GFAP), a glial marker, and ionized calcium binding adapter molecule 1 (Iba1), a marker of microglia. As expected, we observed no colocalization (*Figure 1—figure supplement 1A,B*). We also confirmed that there were no YFP+ cells in bone marrow isolated from *Viaat*-Cre;ROSA-YFP mice (*Figure 1—figure supplement 1C*). To re-express *Mecp2* specifically in *Viaat*-expressing cells, we crossed male *Viaat*-Cre mice with females carrying a *Mecp2* allele with a floxed STOP cassette between the second and third exons (*Mecp2*$^{lox-Stop/Y}$) (*Figure 1A*). Without recombination, only a truncated, non-functioning version of *Mecp2* is expressed, and the resulting mice phenocopy the traditional *Mecp2* null (*Guy et al., 2007*). In the presence of the *Viaat*-Cre allele, however, the STOP cassette is excised and *Mecp2* is transcribed normally in GABAergic neurons, as indicated by colocalization with the GABAergic neuron marker glutamic acid decarboxylase 67 (GAD67, *Figure 1A* left panel); CamKII-positive excitatory neurons (*Figure 1A* right panel) remain null for MeCP2. We characterized males of each of the four genotypes resulting from this cross (wildtype, *Viaat*-Cre, *Mecp2*$^{lox-Stop/Y}$, and *Viaat*-Cre;*Mecp2*$^{lox-Stop/Y}$) for body weight, survival, and the presence of hind limb clasping, tremor, and breathing abnormalities.

The *Viaat*-Cre;*Mecp2*$^{lox-Stop/Y}$ male mice (referred to hereafter as the 'rescue mice') appeared grossly indistinguishable from their wildtype and *Viaat*-Cre littermates (*Figure 1A*). While *Mecp2*$^{lox-Stop/Y}$ mice became obese by seven weeks of life, the body weight of the rescue mice paralleled that of their wildtype and *Viaat*-Cre littermates (*Figure 1B*). Male rescue mice lived significantly longer than their *Mecp2*$^{lox-Stop/Y}$ counterparts: no *Mecp2*$^{lox-Stop/Y}$ male mouse survived past 30 weeks of life, but 47% of the rescue mice survived past one year (22 out of 47 total rescue mice recorded)

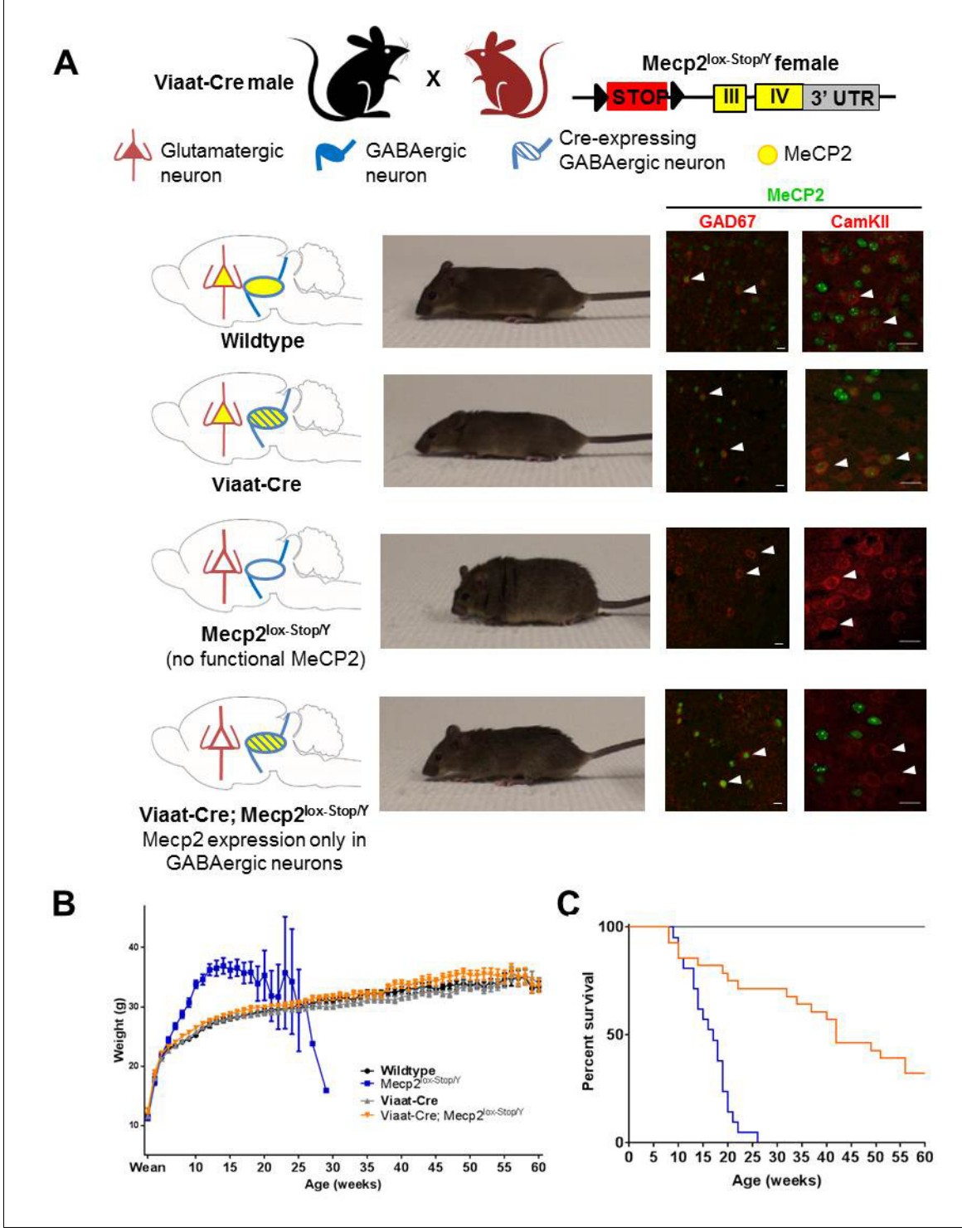

**Figure 1.** Conditional restoration of *Mecp2* expression solely in GABAergic neurons normalized body weight and appearance and significantly extended lifespan. (**A**) Schematic of breeding and mice generated (left column). Photos of each genotype show male *Viaat-Cre; Mecp2^lox-Stop/Y* mice were indistinguishable from wildtype mice at 4 months of age (second column). Cell images show expression of MeCP2 (green) was limited to GAD67-positive GABAergic neurons (red; third column, arrowheads) and was not found in CamKII-expressing excitatory neurons (red, right column, arrowheads). (**B**) Male rescue mice maintained similar body weights to wildtype males throughout life. (**C**) Male rescue mouse lifespan was significantly extended, with ~40% surviving past 1 year of age. n = 33–47 per genotype. Scale bar is 20 μm.

The following figure supplements are available for figure 1:

*Figure 1 continued on next page*

*Figure 1 continued*

**Figure supplement 1.** *Viaat*-Cre was not expressed in glia, microglia, or bone marrow.

**Figure supplement 2.** Male rescue mice maintained more of their body weight up to the point of death and showed improved EEG activity.

(*Figure 1C*). Two rescue mice reached two years of age, which to our knowledge have never been reported with any manipulation of this model. Rescue mice also maintained their body weight throughout life, unlike *Mecp2*<sup>lox-Stop/Y</sup> males, which tended to lose a significant percentage of their body weight in the weeks preceding death (*Figure 1—figure supplement 2A*). We did not observe behavioral seizures in any rescue mouse at any age; using electroencephalogram (EEG) recordings we found that five rescue mice had normal activity patterns while one exhibited abnormal electrographic discharges (*Figure 1—figure supplement 2B*).

## Male GABAergic rescue mice show improvement of multiple RTT-like phenotypes

*Mecp2*-null male mice develop pronounced ataxia early in life (*Chen et al., 2001*; *Guy et al., 2001*), as do the *Mecp2*$^{lox-Stop/Y}$ mice (*Guy et al., 2007*) and the GABAergic-specific *Mecp2* knockout mice (*Chao et al., 2010*). We therefore assessed motor coordination at six weeks of age using the accelerating rotarod and grip strength assays. The *Mecp2*$^{lox-Stop/Y}$ mice had a short latency to fall off the rotarod and impaired grip, as expected, but the male rescue mice performed nearly as well as their wildtype and *Viaat*-Cre littermates in both assays (*Figure 2A,B*). At nine weeks of age, the male rescue mice were not more active than the nulls in the open field assay (OFA), (*Figure 2C*). *Mecp2*$^{lox-Stop/Y}$ mice had noticeable deficits in purposeful forepaw movements: they were unable to build nests at 8 weeks of age or to bury marbles at 9 weeks of age (*Figure 2D–E*). This apraxia-like behavior was completely eliminated in the rescue mice (*Figure 2D–E*). The rescue mice also never developed hindlimb clasping, a sign of motor dysfunction exhibited by both the *Mecp2*$^{lox-Stop/Y}$ mice (*Guy et al., 2007*) and the GABAergic-specific *Mecp2* knockout (*Chao et al., 2010*). Rescue mice also did not display the hypersociability of the *Mecp2*$^{lox-Stop/Y}$ male mice in the partition test at 8 weeks of age (*Figure 2F*).

Not all features of the *Mecp2*$^{lox-Stop/Y}$ mice were rescued, however. Rescue mice developed tremor and labored breathing similar in timing and intensity to the *Mecp2*$^{lox-Stop/Y}$ cohort, suggesting that these symptoms might not be solely dependent on *Mecp2*'s role in inhibitory signaling. This indicates that the development of tremor is particularly dependent on proper excitatory network activity; indeed, loss of *Mecp2* from GABAergic neurons does not result in tremor (*Chao et al., 2010*), while glutamatergic conditional knockout mice do develop a tremor which is not observed in glutamatergic conditional rescue mice (see companion paper *Meng et al., 2016*). Breathing, however, seems to be affected by dysfunction in both excitatory and inhibitory networks, as GABAergic conditional *Mecp2*-knockout mice do develop abnormal breathing patterns (*Chao et al., 2010*). In addition, the rescue mice shared the acoustic startle deficit observed in the *Mecp2*$^{lox-Stop/Y}$ mice (*Figure 2G*), with no differences in prepulse inhibition at any decibel level noted (data not shown). Interestingly, rescue mice also showed a partial benefit to anxiety: *Mecp2*$^{lox-Stop/Y}$ and rescue mice performed similarly in the elevated plus maze test (*Figure 2H*), but the rescue mice were indistinguishable from wildtype and *Viaat*-Cre control mice in the light-dark box test *Figure 2I*. Taken together, these findings indicate that regulation of startle and tremor is more reliant on proper excitatory signaling than on inhibition, while anxiety may be partially mitigated by improved GABAergic function that in turn somewhat normalizes glutamatergic circuit function.

As the majority of the rescue male mice outlived their *Mecp2*$^{lox-Stop/Y}$ littermates, we were able to assess whether these mice maintained the observed behavioral rescue later in life. We tested wildtype, *Viaat*-Cre, and rescue mice at 30 weeks of age for motor deficits by the open field assay, accelerating rotarod, and grip strength meter. By 30 weeks, all *Mecp2*$^{lox-Stop/Y}$ mice had died, while over 60% of the rescue mice were still alive (*Figure 1C*). The surviving rescue mice appeared grossly similar to the wildtype and *Viaat*-Cre male mice (*Figure 3A*) but displayed an increased breathing rate and noticeable tremor. The older rescue mice tended to move less

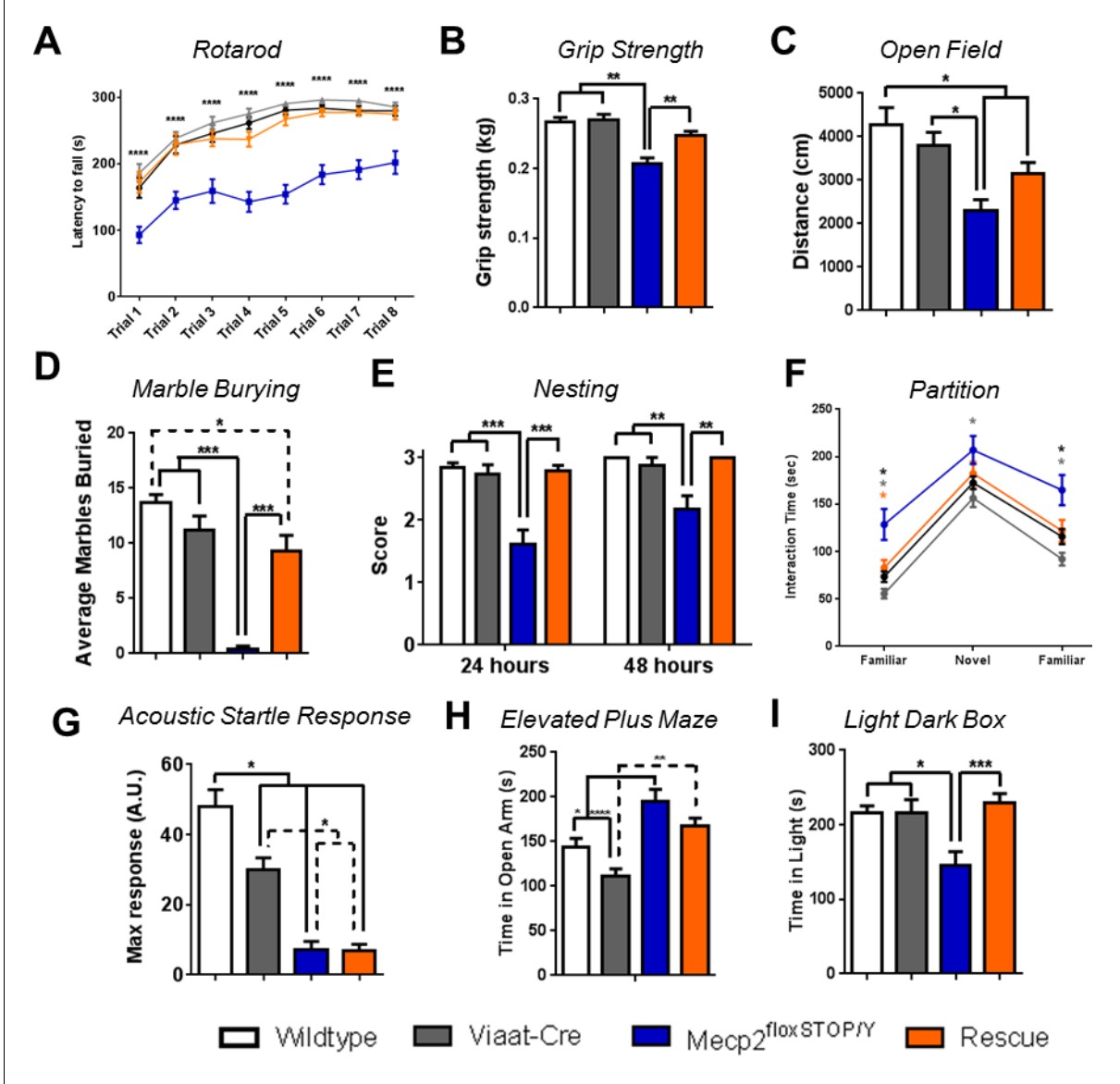

**Figure 2.** Male rescue mice exhibited significant behavioral rescue. (A–B) Male rescue mice were indistinguishable from wildtype and *Viaat*-Cre controls at 6 weeks of age on the rotarod (n = 23–28 per genotype) (A) and grip strength (B) assays (n = 20–26 per genotype). (C) Rescue mice trended toward improved locomotor activity in the open field assay at 9 weeks of age (n = 8–15 per genotype). (D–E) Apraxia at 9 weeks of age was reversed in rescue mice as indicated by marble burying (n = 9–15) (D) or, at 8 weeks, by nest building (E, n = 23–27 per genotype). (F) Rescue mice had similar sociability to wildtype mice in the partition test at 8 weeks of age. * represents significance of genotype (by color) compared to *Mecp2*[lox-Stop/Y]. (G) Acoustic startle response was unchanged in rescue mice at 8 weeks of age (n = 16–21). (H−I) Rescue mice were similar in anxiety behavior to *Mecp2*[lox-Stop/Y] in the elevated plus maze test (G) but were similar to wildtype in the light dark box assay (I). Error bars show SEM. *p<0.05 **p<0.01 ***p<0.001

than their wildtype and *Viaat*-Cre counterparts (***Figure 3B***), were slightly less coordinated on the rotarod (***Figure 3C***), and slightly weaker in their grip strength (***Figure 3D***). Rotarod performance and grip strength were impaired only when compared to those of the *Viaat*-Cre control. This apparent deficit, however, was likely due to the variable health of the rescue mice, some of which were more ill than others at 30 weeks. Regardless, as a group the rescue mice were able to perform near wildtype levels, and overall the behavioral rescue produced by *Mecp2* reexpression in GABAergic neurons persisted later in life.

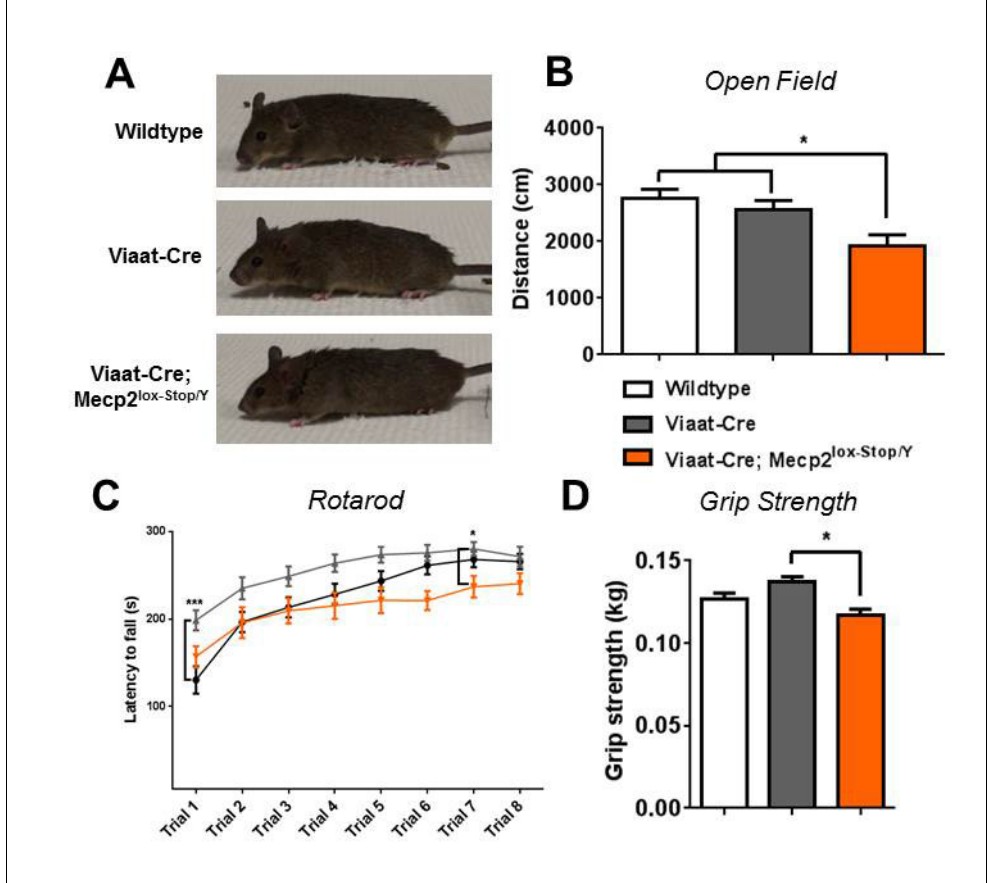

**Figure 3.** Rescue persisted in male mice at 30 weeks of age. (**A**) Male rescue mice at 30 weeks of age continued to appear indistinguishable from wildtype and *Viaat*-Cre littermates. (**B**) Locomotion in the open field assay decreased when compared to wildtype but not *Viaat*-Cre. (**C–D**) Rotarod performance was maintained at wildtype levels, as was grip strength (**D**). n = 20–25 per genotype. Error bars show SEM. *p<0.05

## Behavioral rescue correlates with normalization of GABA levels and gene expression

Loss of MeCP2 leads to decreases in GABA concentrations (*Chao et al., 2010*) and widespread gene expression changes (*Chahrour et al., 2008*); normalization of these deficits may underlie the behavioral rescue we observe. Indeed, we found that GABA concentrations in the striata of 8-week-old *Mecp2*[lox-Stop/Y] mice were significantly lower than in wildtype but were restored to wildtype levels in the rescue mice (*Figure 4A*). We then assessed changes in the expression of genes known to be involved in neuronal GABAergic function using tissue from the cerebella of 8-week-old mice. Interestingly, levels of *Gad1*, *Gad2*, and *Vgat* were all decreased in the *Mecp2*[lox-Stop/Y] but were identical to wildtype and *Viaat*-Cre levels in the rescue mice (*Figure 4B*). We next assessed a slate of genes that are expressed in inhibitory neurons and known to be altered in *Mecp2*-null brains and observed the expected downregulation of *Nxph4*, *Gabra3*, *Kcng4*, *Opn3*, and *Scg2* as well as upregulation of *A1593442*, *Robo2*, *Rassf8*, and *Cabp7* in the *Mecp2*[lox-Stop/Y] cerebellum. Tissues from rescue mice, however, showed reversal of these expression patterns, with the exception of *Robo2*, which trended to normalization (*Figure 4C*). Re-expression of *Mecp2* in GABAergic neurons thus normalizes several transcriptional changes related to inhibitory neuron function.

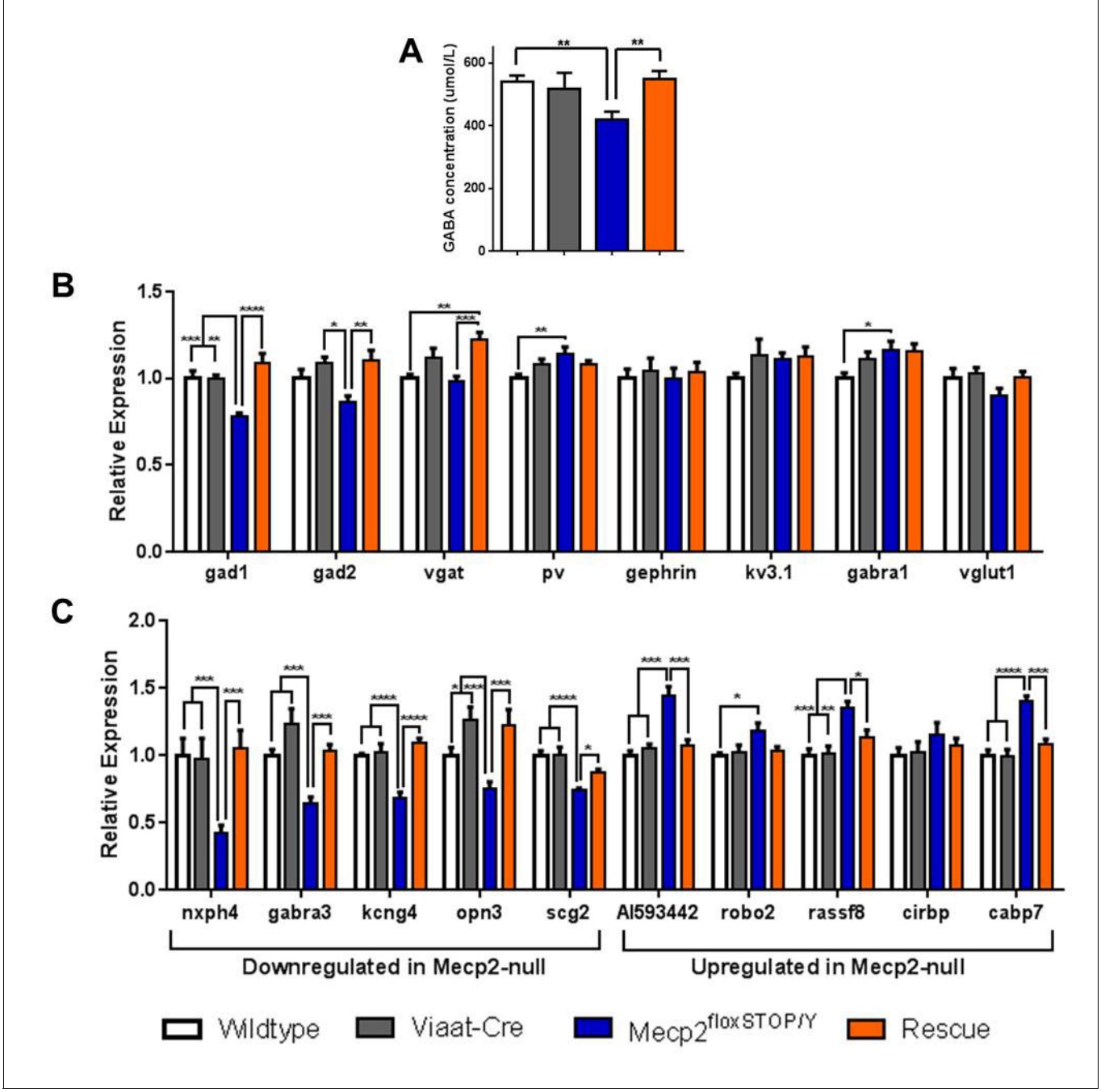

**Figure 4.** Expression of *Mecp2* in GABAergic neurons normalizes levels of GABA and other MeCP2 responsive genes. (**A**) GABA concentrations were restored to wildtype levels in striatum from rescue mice (n = 4–5 per genotype). (**B**) RNA expression of genes related to GABAergic neuronal function, particularly *gad1* and *gad2*, were normalized in rescue mouse cerebella. (**C**) Genes known to be downregulated or upregulated in *Mecp2*-null cerebellum were normalized in the rescue mice. Error bars show SEM. *p< 0.05 **p<0.01 ***p<0.001 ****p<0.0001. n = 6–7 per genotype.

## Expression of *Mecp2* in GABAergic neurons improves cortical neuronal excitability

To understand the cellular mechanism of the observed behavioral rescue, we recorded miniature inhibitory postsynaptic currents (mIPSCs) and miniature excitatory postsynaptic currents (mEPSCs) from 5–6-week old wildtype, *Viaat*-Cre, *Mecp2*^lox-Stop/Y and rescue male mice (n = 3 mice per genotype). We prepared cortical slices and conducted whole cell recordings on pyramidal neurons in

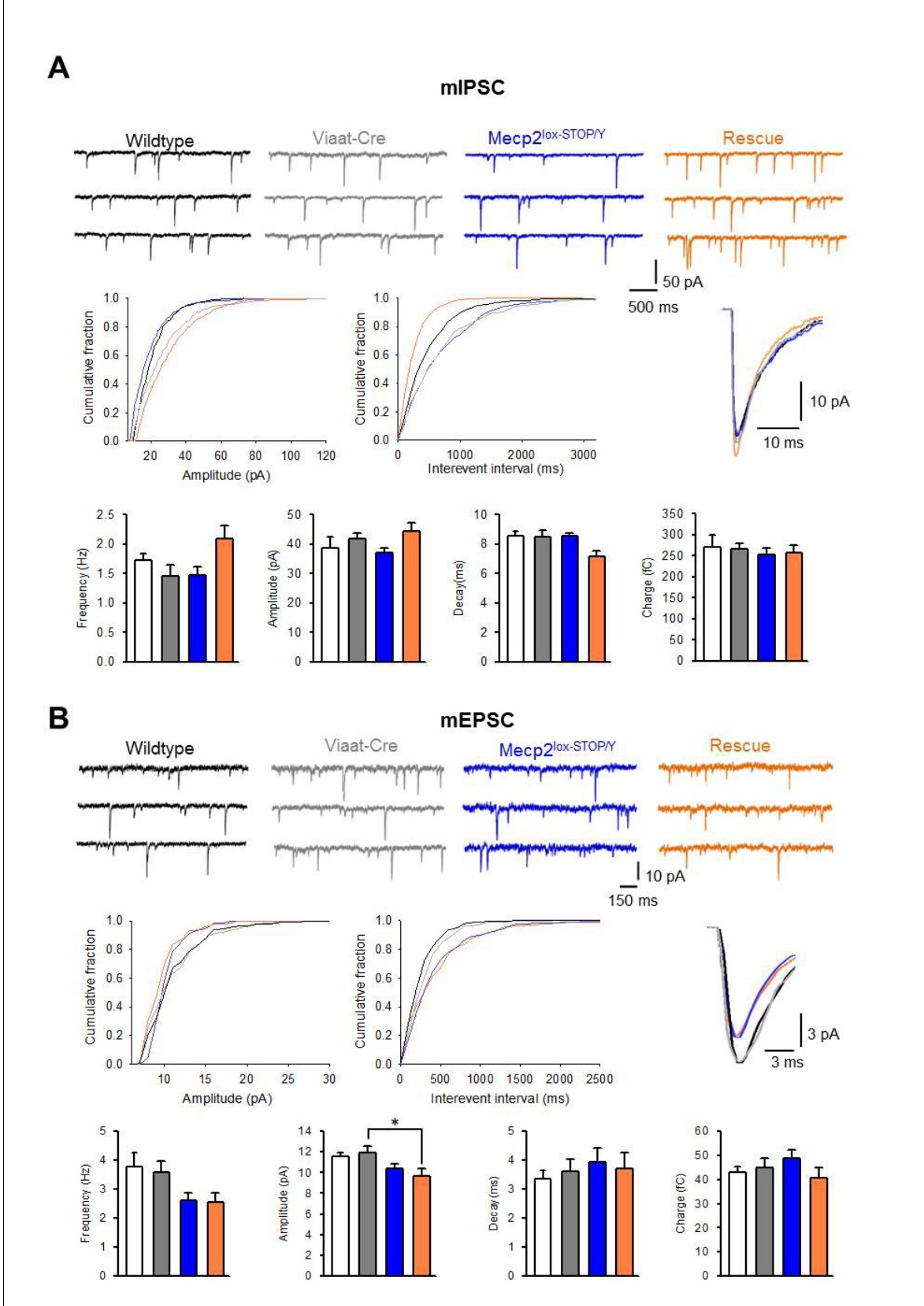

**Figure 5.** Inhibitory signaling in rescue mice showed some improvement while excitatory signaling was unchanged. (**A–B**) Sample traces of mIPSC (A, Wildtype = 11, *Viaat*-Cre = 17, *Mecp2*[lox-STOP/Y] = 14, Rescue = 11) and mEPSC (B, Wildtype = 17, *Viaat*-Cre = 16, *Mecp2*[lox-STOP/Y] = 18, Rescue = 17)
*Figure 5 continued on next page*

*Figure 5 continued*

from pyramidal cells of the somatosensory cortex of wildtype, *Viaat*-Cre, *Mecp2*$^{lox-STOP/Y}$, and Rescue male mice, including cumulative distributions of amplitude and interval, grand average minis, and summaries of frequency, amplitude, decay, and average charge. Error bars show SEM. *p<0.05

The following figure supplement is available for figure 5:

**Figure supplement 1.** Inhibitory synapse numbers and spontaneous action potential firing are normalized in rescue mice.

layer II/III of the somatosensory cortex, a region that showed alterations in these currents in the GABAergic conditional knockout mouse (*Chao et al., 2010*). We found that the frequency and amplitude of mIPSCs in the rescue mice showed a strong trend to increase when compared to the other genotypes, while the decay rate trended lower (*Figure 5A*), although none of these changes reached statistical significance when analyzed using a one-way ANOVA with a Tukey posthoc test (*Supplementary file 1*). Taken together, these data suggest a trend to increased presynaptic GABA release with *Mecp2* reexpression in GABAergic neurons. Furthermore, rescue mice had similar numbers of vesicular GABA transporter (VGAT)-positive puncta to wildtype and *Viaat*-Cre controls in the hippocampal CA1, while *Mecp2*$^{lox-Stop/Y}$ mice had significantly fewer VGAT+ synapses, indicating that reexpression of *Mecp2* in GABAergic neurons normalizes inhibitory synapse number (*Figure 5—figure supplement 1A*). In contrast, the frequency and amplitude of mEPSCs in rescue mice were lower than those of wildtype and *Viaat*-Cre controls but similar to those of *Mecp2*$^{lox-Stop/Y}$ mice (*Figure 5B*), indicating that *Mecp2* reexpression in GABAergic neurons failed to rescue mEPSCs. Vesicular glutamate transporter (Vglut1)-positive puncta numbers in the CA1 were not significantly different across genotypes, although there was a strong trend to fewer Vglut1+ puncta in the rescue mice (*Figure 5—figure supplement 1A*). Expression of *Mecp2* solely in GABAergic neurons thus appears to mildly enhance inhibitory neuronal signaling but has no effect on excitatory synaptic activity.

To determine if this enhancement has any effect on neuronal firing activity, we recorded spontaneous action potentials from layer V pyramidal neurons of the somatosensory cortex of 6–8 week old male wildtype, *Viaat*-Cre, *Mecp2*-null, and rescue mice. While *Mecp2*-null mice exhibited significantly fewer spontaneous action potentials than wildtype animals, rescue mice showed normalized activity (*Figure 5—figure supplement 1B*). Intrinsic neuronal excitability, assessed by intracellular injection of step currents, did not vary among genotypes (*Figure 5—figure supplement 1C*). It is thus clear that reexpression of *Mecp2* in GABAergic neurons normalizes neuronal firing activity in the cortex; taken together with the improved inhibitory input, this normalization could feasibly be due to normalized upstream disinhibition.

## RTT-like behaviors are significantly improved in female rescue mice

Most mouse models of RTT rely on male mice to avoid the confounding influence of X chromosome inactivation. Given that RTT primarily affects females, however, we wanted to determine whether the reexpression of *Mecp2* in GABAergic neurons could exert similar benefits in female *Mecp2*$^{lox-Stop/+}$ mice. We characterized females generated from the same cross that produced the *Mecp2*$^{lox-Stop/Y}$ male mice (*Figure 1A*). All genotypes from this cross lived normal lifespans, with no premature death noted in either *Mecp2*$^{lox-Stop/+}$ or rescue females. By 15 weeks of age, the female *Mecp2*$^{lox-Stop/+}$ mice, which lack functional MeCP2 in roughly half their cells, were noticeably heavier than their wildtype and *Viaat*-Cre female littermates and developed a readily observable tremor and labored breathing (*Figure 6A,B*).

We assessed behavior in the female mice at later ages than the male mice, as the effects of *Mecp2* deletion are delayed in the female *Mecp2* heterozygous mice (*Samaco et al., 2013*). Rotarod performance was impaired in the female *Mecp2*$^{lox-Stop/+}$ mice by 9 weeks of age (*Figure 6C*), although there was no difference in locomotion in the open field assay (*Figure 6—figure supplement 1A*). In contrast, the rescue females were only slightly heavier than wildtype females (*Figure 6A,B*), and they performed as well as wildtype on the rotarod at 9 weeks of age (*Figure 6C*). We also observed rescue of ataxic phenotypes at 18 weeks of age; while there was no significant difference in locomotion in the open field, the female rescue mice were no different from wildtype females in the footslip assay (*Figure 6—figure supplement 1B,C*). Like the male rescue mice,

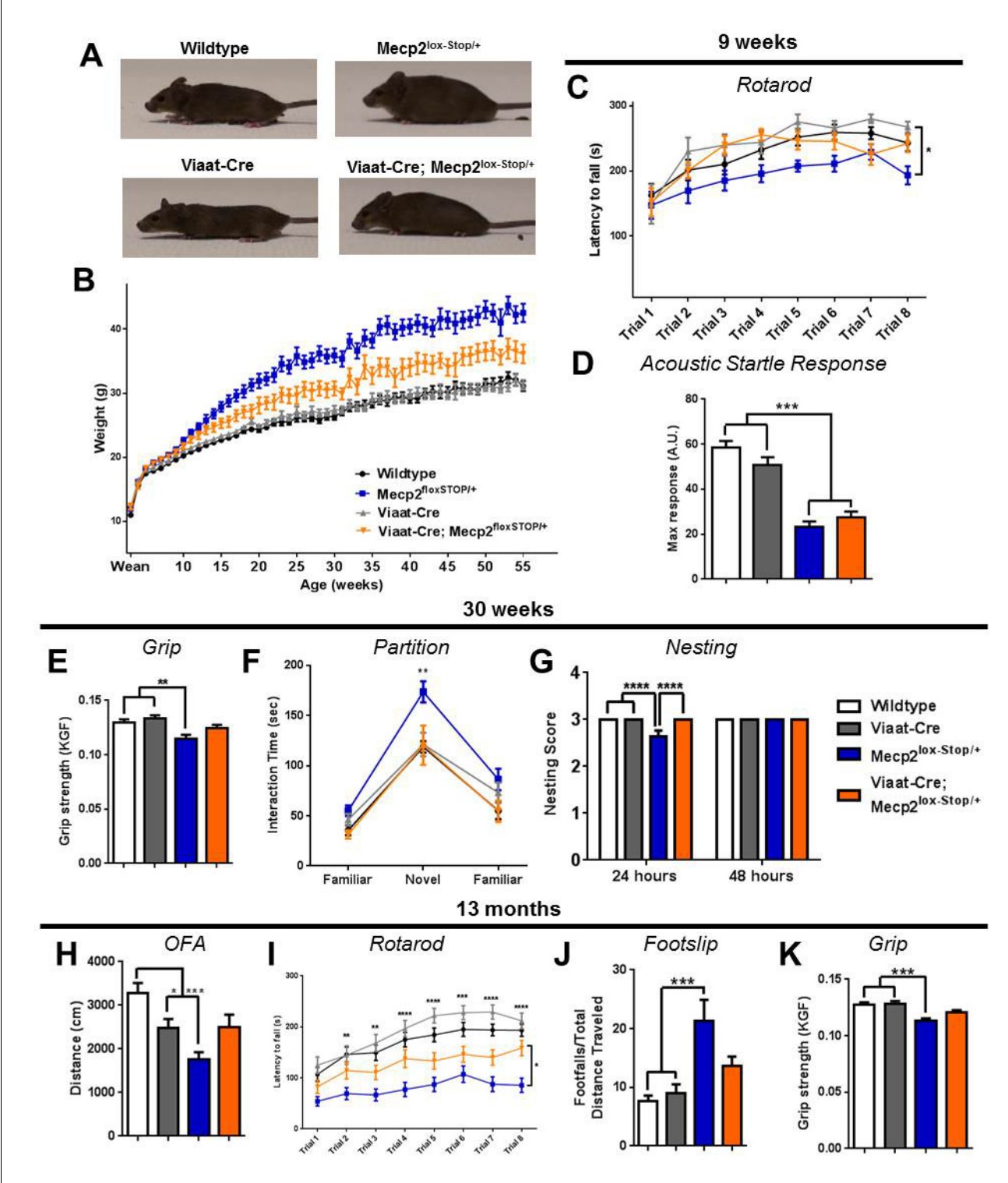

**Figure 6.** Female rescue mice exhibited a partial but sustained functional recovery. (A) Female rescue mice appeared grossly normal at 23 weeks of age. (B) Female rescue mice body weight was partially rescued (n = 27–37 per genotype). (C) Female rescue mice performed better at 9 weeks of age on the accelerating rotarod than $Mecp2^{lox-STOP/+}$ females, but not to wildtype levels (n = 14–16 per genotype). (D) Acoustic startle response at 9 weeks of age was not rescued (n = 14–17 per genotype). (E–G) At 30 weeks of age, female rescue mice showed partial rescue of grip strength (E) and complete rescue of hypersociability in the partition test (F) and nesting ability (n = 13–18 per genotype) (G). (H–K) Rescue females exhibited a partial but sustained rescue of locomotion in the open field (H), rotarod (I), footslip (J), and grip strength (K) assays (n = 20–35 per genotype). Error bars show SEM. *p<0.05 **p<0.01 ***p<0.001 ****p<0.0001.

*Figure 6 continued on next page*

*Figure 6 continued*

The following figure supplement is available for figure 6:

**Figure supplement 1.** Female rescue mice show partial rescue particularly at late ages.

however, the startle deficit of the female rescue mice at nine weeks was unaffected by reexpression of *Mecp2* in inhibitory neurons (*Figure 6D*), and we observed no difference in prepulse inhibition (data not shown); in contrast, this startle deficit was rescued in the glutamatergic neuron rescue female mice (see companion paper *Meng et al., 2016*).

The male *Mecp2*[lox-Stop/Y] phenotype is severe, killing the mouse within the first five months of life, but female *Mecp2*[lox-Stop/+] mice commonly live past the one-year mark. This mimics the human disease, as some female RTT patients live full lifespans with considerable supportive care. To determine if the behavioral rescue observed at younger ages was sustained later in life, we assessed behavior in females at 30 weeks and 13 months of age. At both ages, *Mecp2*[lox-Stop/+] females displayed a pronounced tremor and hindlimb clasping when suspended. At 30 weeks of age, female *Mecp2*[lox-Stop/+] mice had similar locomotion to the other genotypes (*Figure 6—figure supplement 1D*), but they had poor grip strength (*Figure 6E*) and impaired dowel walk performance (*Figure 6—figure supplement 1E*). In addition, they were hypersocial when presented with a novel partner mouse in the partition test (*Figure 6F*) and were slower to build nests (*Figure 6G*). In contrast, the female rescue mice performed similarly to wildtype mice in each of these assays, although the difference from the *Mecp2*[lox-Stop/+] mice was not statistically significant at early ages because of the latter's relatively mild impairment (*Figure 6E–G*, *Figure 6—figure supplement 1D–E*). At 13 months of age, female rescue mice moved noticeably more than the *Mecp2*[lox-Stop/+] females (*Figure 6H*), were less ataxic by rotarod (*Figure 6I*), footslip (*Figure 6J*), and dowel walk (*Figure 6—figure supplement 1F*), and had better grip strength (*Figure 6K*). On each behavioral metric the performance of female rescue mice at this late age was worse than wildtype but still better than that of *Mecp2*[lox-Stop/+] mice; interestingly, reexpression of *Mecp2* solely in glutamatergic neurons resulted in a near complete reversal of RTT-like phenotypes (see companion paper *Meng et al., 2016*), possibly because restoring glutamatergic function in a brain with approximately half of the GABAergic neurons retaining *Mecp2* expression due to random X inactivation more closely approximated a normal excitatory-inhibitory balance. Notably, the GABAergic rescue mice had an observable tremor, but they never developed the hindlimb-clasping characteristic of the *Mecp2*[lox-Stop/+] mice. The mitigation of the RTT-like phenotype observed in female rescue mice thus persisted well into their second year of life.

## Discussion

Perturbations in the balance between excitatory and inhibitory signaling have been hypothesized to underlie several neuropsychiatric disorders, including autism (*Vattikuti and Chow, 2010*; *Gogolla et al., 2009*; *Rubenstein and Merzenich, 2003*), schizophrenia (*Belforte et al., 2010*), and RTT (*Dani et al., 2005*). Artificial elevation of the ratio between excitatory and inhibitory signaling leads to social deficits in mice, an effect that is rescued by an increase of inhibitory circuit activity (*Yizhar et al., 2011*). Here, we present evidence that increasing inhibitory activity alone can reverse the majority of symptoms of a mouse model of RTT. Using existing, well-characterized mouse lines, we restored full-length *Mecp2* transcription solely in GABAergic neurons and enhanced inhibitory function in an otherwise *Mecp2*-null brain. This was sufficient to extend lifespan significantly and rescue body weight, ataxia, and apraxia in both male and female rescue mice.

At the time of this writing, there have been several conditional knockouts of *Mecp2* that target different neuronal subtypes, each replicating a few aspects of the *Mecp2* deletion phenotype: *Sim1* conditional knockout mice are hyperphagic and become obese (*Fyffe et al., 2008*); mice with *Mecp2* deleted from dopaminergic neurons develop some motor deficits (*Samaco et al., 2009*); and conditional knockout of *Mecp2* from basal forebrain cholinergic neurons may result in

decreased anxiety and impaired social interaction (*Zhang et al., 2016*). Deletion of *Mecp2* from two subtypes of inhibitory neurons, PV- and SOM-expressing neurons, also shortened lifespan and replicated a subset of the phenotypes of global *Mecp2* loss: PV-conditional knockout mice lacked critical period plasticity in the visual cortex (*He et al., 2014*) and developed motor, learning and memory, and social interaction deficits, while SOM-conditional knockout mice exhibited stereotyped behaviors and seizures (*Ito-Ishida et al., 2015*); interestingly, reexpression of *Mecp2* in these individual populations prolongs life and improves disease scores and auditory information processing (*Goffin et al., 2014*). Of all the conditional knockouts, however, the GABAergic conditional knockout mouse most closely replicates the effect of global *Mecp2* deletion, with premature lethality, motor dysfunction, learning and memory deficits, and stereotypies. Only the anxiety and the persistent tremor of the *Mecp2*-null mouse is not reproduced (*Chao et al., 2010*)—behaviors that are reproduced instead by the glutamatergic conditional knockout, along with motor incoordination, acoustic startle deficits, and early lethality (see companion paper *Meng et al., 2016*). In parallel, our rescue of *Mecp2* expression solely in GABAergic neurons in otherwise *Mecp2*-null male mice completely reverses hypersociability, ataxia, and apraxia, as well as partially normalizing anxiety, but has no effect on acoustic startle responses or tremor. Meng et al's glutamatergic rescue of *Mecp2* expression in the same *Mecp2*-null male mice completely normalizes anxiety, tremor, and acoustic startle deficits, as well as reversing hypersociability and extending lifespan, but does not affect motor coordination. This extraordinary complementarity makes it clear that while certain behaviors may be mainly dependent on the proper function of only excitatory neurons (ie acoustic startle response, tremor) or only inhibitory neurons (ataxia), other behaviors are dependent on both populations functioning properly (social behavior). However, caution must be exercised in assessing the exact contributions of each population, as the *Viaat*-Cre allele also expresses in glycinergic neurons, which are important for breathing and proper brainstem function (*Rahman et al., 2015*). Furthermore, small populations of neurons in the lateral and medial habenula, the auditory cortex, and possibly in the CA3 release both GABA and glutamate (*Shabel et al., 2014*; *Root et al., 2014*; *Safiulina, 2006*; *Noh et al., 2010*; *Uchigashima et al., 2007*), which further complicates the picture. However, inhibitory neurons are found throughout the brain and, due to their exquisite sensitivity to the surrounding circuitry, can influence the behavior of large populations of surrounding neurons (*Xue et al., 2014*; *Bonifazi et al., 2009*), while glycinergic and GABA/glutamate-coreleasing neurons are far smaller in number and restricted to discrete regions of the brain. It is thus reasonable to attribute the bulk of the behavioral rescue to the improved inhibitory function we observe in the rescue mice. Thus, the work presented here not only indicates that some behavioral aspects of RTT pathogenesis are modularly regulated by specific populations of neurons but also emphasizes the need for a complete understanding of the interplay between excitatory and inhibitory neurons in behavioral outputs.

It is telling that neither glutamatergic nor GABAergic rescue of *Mecp2* completely rescued premature lethality, although both rescue mice displayed similarly extended lifespans (*Figure 1C*, also ). A normal lifespan in the male mice is likely particularly dependent on the function of both neuron types. In addition, there may also be a role for glia in RTT pathogenesis, as these cells are critical for neurotransmitter recycling (*Marcaggi and Attwell, 2004*). Notably, reactivation of *Mecp2* transcription solely in GFAP-positive astrocytes rescues the respiratory abnormalities of the *Mecp2*-null mouse and extends survival (*Lioy et al., 2011*), whereas our GABAergic rescue mouse showed no clear improvement in breathing. It is therefore conceivable that normal respiration and normal lifespan depend on an interplay between proper excitation, inhibition, and astrocyte function.

Differences between male and female mice in the GABAergic rescue responses are worth some consideration. In the male rescue mice, inhibitory cells express *Mecp2*, but all excitatory neurons are null for *Mecp2*. When *Mecp2* expression is restored in inhibitory neurons in males, the increase in GABA signaling may effectively counteract much of the deficit resulting from the loss of *Mecp2* in all excitatory neurons, possibly explaining the profound rescue of disease-related symptoms in the male mice. In the female rescue mice, however, random X-inactivation should allow roughly 50% of the excitatory and inhibitory neurons to retain functional *Mecp2* (*Young and Zoghbi, 2004*), resulting in a heterogeneity of excitatory neuron activity within the brain of female mice that may neutralize much of the benefit gained by normalizing inhibitory neuron function. This idea is supported by the work of Meng et al., who show that rescue of *Mecp2* expression solely in excitatory neurons confers greater behavioral rescue in females than males. These glutamatergic rescue females benefit

from a fully normalized excitatory neuron population as well as normal functioning from approximately 50% of inhibitory neurons, resulting in a strong rescue that is less impeded by heterogeneity. Males, however, are still left with dysfunctional inhibitory circuits and thus experience less benefit from restoration of MeCP2 levels in excitatory neurons. It is clear that inhibitory neurons are a valuable target for future pharmacological studies; future therapies that boost GABAergic function in both wildtype and *Mecp2*-null cells might prove more effective than a treatment targeted to the null cells alone. However, the work presented here and the work of Meng et al. also indicate that to achieve complete reversal of RTT symptoms, it may be necessary to pharmacologically elevate both inhibition and excitation.

Lastly, the discovery that genetic restoration of *Mecp2* expression solely in GABAergic neurons improves several phenotypes in both male and female mice throughout their lifespans renders this pathway a viable route for modifying disease course, even if only in the first few months to years of the disease. Pharmacological therapies enhancing GABAergic signaling might confer noticeable benefit to RTT syndrome patients, buying time until additional therapies can be added to modulate other pathways such as glutamatergic signaling. However, our findings suggest great care is warranted in the choice of disease model for any future drug trial, as the *Mecp2*-null male and *Mecp2*-heterozygous female mice responded very differently to genetic rescue of *Mecp2* expression. For a therapeutic to be truly translatable to human RTT patients, it must be tested in both models.

## Materials and methods

### Mouse husbandry and handling

All mouse care and manipulation was approved by the Baylor College of Medicine Institutional Animal Care and Use Committee (IACUC, Protocol AN-1013). Mice were housed in an AAALAS-certified Level 3 facility on a 14 hr light cycle. Male FVB mice carrying the *Viaat*-Cre transgenic line (*Chao et al., 2010*) were mated with 129S6SvEvTac females heterozygous for the *Mecp2*[lox-Stop] allele (*Guy et al., 2007*), resulting in male and female F1 hybrid offspring. For the reporter experiments, *Viaat*-Cre males were crossed with female C57Bl/6 ROSA-YFP mice. For the spontaneous firing experiments, *Mecp2*[-/Y] male mice from the same hybrid background were used. After weaning, all mice were group housed (2–5 mice per cage) as a mix of genotypes. All mice included in the survival curve were weighed weekly and scored according to the 6-category disease scoring scale, as previously described (*Guy et al., 2007*) (Wildtype = 41, *Viaat*-Cre = 37, *Mecp2*[lox-Stop] = 33, *Viaat*-Cre; *Mecp2*[lox-Stop] = 47). The investigator remained blind to the genotypes of all tested mice during phenotypic characterization and behavioral testing.

### Behavioral tests

Male mice used for behavioral assays were divided into 2 cohorts, each being assessed by different assays at different time points. All behavioral assays were conducted during the light cycle, generally in the afternoon. For Cohort 1, mice (n = 21–24 per genotype) were tested at 6 weeks of age on OFA, grip strength, and rotarod; at 8 weeks for partition test with nesting assessment, PPI, and conditioned fear, and at 30 weeks by OFA, grip strength, and rotarod. Cohort 2 (n = 9–15 per genotype) was tested at 9 weeks for OFA and marble burying. For females, Cohort 1 (n = 8–18 per genotype) was tested at 18 weeks for OFA, grip strength, footslip, and partition with nesting assessment. Cohort 2 (n = 11–17 per genotype) was tested at 9 weeks for OFA, rotarod, PPI, and conditioned fear, at 30 weeks for OFA, grip strength, rotarod, partition with nesting assessment, and marble burying. Both female cohorts were tested at 1 year for OFA, grip strength, rotarod, footslip, and conditioned fear. All mice were assessed weekly for body weight and disease score, which included movement, gait, hindlimb clasping, tremor, breathing, and general condition (*Guy et al., 2007*). The investigator was blinded to all genotypes after completion of data collection.

### Open field assay (OFA)

Mice were habituated for 30 min in the test room lit at 200 lux with white noise playing at 60 dB. Each mouse was placed singly in the open field apparatus (OmniTech Electronics, Columbus, OH) and allowed to move freely for 30 min. Locomotion parameters and zones were recorded using the

Fusion activity monitoring software. Data is shown as mean ± standard error of mean and was analyzed by one-way ANOVA with Tukey's post hoc analysis.

### Elevated plus maze (EPM)

Mice were habituated for 30 min in the test room lit at 200 lux with white noise playing at 60 dB. The elevated plus maze is a plus sign-shaped maze with two opposite arms enclosed by walls and two opposite arms open without walls. The entire maze is elevated above the floor. Mice were placed singly at the intersection of the four arms and allowed to move freely for 10 min. Activity was recorded by a suspended digital camera and recorded by the ANY-maze software (Stoelting Co., Wood Dale, IL). Data is shown as mean ± standard error of mean. Time and distance in the open arm were each analyzed by one-way ANOVA with Tukey's post hoc analysis.

### Light/dark box

Mice were habituated for 30 min in the test room lit at 200 lux with white noise playing at 60 dB. Mice were placed singly in the light side of the light dark apparatus (Omnitech Electronics) and allowed to move freely for 10 min. Locomotion parameters and zones were recorded using Fusion activity monitoring software. Data is shown as mean ± standard error of mean. Time in Light was analyzed by one-way ANOVA with Tukey's post hoc analysis.

### Grip strength

Mice were habituated in the test room for 30 min. Each mouse was allowed to grab the bar of a digital grip strength meter (Columbus Instruments, Columbus, OH) with both forepaws while being held by the tail and then pulled away from the meter with a constant slow force until the forepaws released. The grip (in kg of force) was recorded and the procedure repeated twice for a total of three pulls, which were averaged for the final result. Data is shown as mean ± standard error of mean. Grip strength was analyzed by one-way ANOVA with Tukey's post hoc analysis.

### Dowel walk

Mice were habituated in the test room for 30 min. The mouse was placed on a 0.635 cm diameter dowel with all four paws allowed to grip the dowel. Latency to fall and number of side touches were recorded during the two minute test. Data is shown as mean ± standard error of mean and was analyzed by one-way ANOVA with Tukey's post hoc analysis.

### Rotarod

Mice were placed on the rotating cylinder of an accelerating rotarod apparatus (Ugo Basile, Varese, Italy) and allowed to move freely as the rotation increased from 5 rpm to 40 rpm over a five-minute period. Latency to fall was recorded when the mouse fell from the rod or when the mouse had ridden the rotating rod for two revolutions without regaining control. Data is shown as mean ± standard error of mean. Latency to fall was analyzed by two-way ANOVA with Bonferroni's post hoc analysis.

### Footslip

Mice were habituated in the test room for 30 min. Each mouse was placed in a footslip chamber consisting of a plexiglass box with a floor of parallel-positioned rods and allowed to move freely for 10 min. Movement was recorded by a suspended digital camera, while footslips were recorded using ANY-maze software (Stoelting Co.). At the completion of the test, mice were removed to their original home cage. Total footslips were normalized to the distance traveled for data analysis. Data is shown as mean ± standard error of mean and analyzed by one-way ANOVA with Tukey's post hoc analysis.

### Marble burying test

Mice were habituated for 30 min in the test room. A standard mouse housing cage was 50% filled with clean bedding material, and 20 black glass marbles were placed in a 4x5 grid pattern on the surface of the bedding. Mice were placed singly into the prepared cage for 30 min. After the mouse was removed, the number of buried marbles were counted, with a marble considered buried if 75%

of its surface was covered with bedding. Data is shown as mean ± standard error of mean and analyzed by one-way ANOVA with Tukey's post hoc analysis.

## Partition test and nesting assessment

Mice were single-housed for 48 hr on one side of a standard mouse housing cage. The cage was divided across its width by a divider with holes small enough to allow scent but no physical interaction. The test mouse was provided with a KimWipe folded in fourths as nesting material. At 24 hr and 48 hr of single-housing, the KimWipe was assessed for nesting score, as described previously (*Chao et al., 2010*). At least 16 hr before the partition tests, a novel age- and gender-matched partner mouse of a different background was placed on the opposite side of the partition. On the day of the test, the cage was placed on a well-lit flat surface. All nesting material and water bottles were removed from both sides of the cage, and the test mice were observed for 5 min while interaction time with the now-familiar partner mouse was recorded. Interactions involved the test mouse smelling, chewing, or actively exploring the partition. At the end of the first test (Familiar 1), the familiar partner mouse was replaced by a novel mouse of the same age, gender, and strain, and test mouse interactions were recorded for five minutes (Novel). The novel mouse was then removed and the familiar partner mouse returned to the cage, followed by observation for another 5 min (Familiar 2). At the completion of the partition test, test mice were returned to their original home cage. Data is shown as mean ± standard error of mean. Interaction times were analyzed by two-way ANOVA with Bonferroni's post hoc analysis, and nesting scores were analyzed by one-way ANOVA with Tukey's post hoc analysis.

## Prepulse Inhibition (PPI)

Mice were habituated for 30 min outside the test room. Each mouse was placed singly in SR-LAB PPI apparatus (San Diego Instruments, San Diego, CA), which consisted of a Plexiglass tube-shaped holder in a sound-insulated lighted box, with 70 dB white noise and allowed to habituate for 5 min. The mouse was presented with eight types of stimulus, each presented six times in pseudo-random order with a 10–20 ms intertrial period: no sound, a 40 ms 120 db startle burst, three 20 ms prepulse sounds of 74, 78, and 82 dB each presented alone, and a combination of each of the three prepulse intensities presented 100 ms before the 120 dB startle burst. After the test, mice were returned to their home cage. The acoustic startle response was recorded every 1 ms during the 65 ms period following the onset of the startle stimulus and was calculated as the average response to the 120 db startle burst normalized to body weight. Percent prepulse inhibition was calculated using the following formula: (1-(averaged startle response to prepulse before startle stimulus/averaged response to startle stimulus)) x 100. Data are shown as mean ± standard error of mean. Percent prepulse inhibition was analyzed by two-way ANOVA with Bonferroni's post hoc analysis, and acoustic startle response was analyzed by one-way ANOVA with Tukey's post hoc analysis.

## Immunohistochemistry

Mice (5–6 weeks old, n = 2–3) were perfused transcardially first with 1XPBS and then by 4% PFA. The brains were removed and postfixed overnight in 4% PFA, followed by storage in 30% sucrose until sinking. For sectioning, fixed brains were embedded in TissueTek and sectioned by cryostat at 25 um. Sections were maintained in 1XPBS with sodium azide at 4°C until stained using a free-floating protocol. In short, sections were washed once in 1XPBS and then blocked for 1 hr at room temperature (RT) in 3% normal goat serum (NGS) with 0.3% TritonX in 1XPBS. Primary antibodies were diluted in 1XPBS with 1.5% NGS with 0.3% Tween-20 in 1XPBS, and sections were stained for 48 hr shaking at 4°C in a ~300 µl working volume. Primary antibodies used were: chicken anti-GFP (Abcam, Cambridge UK, Cat# ab13970) 1:2000; mouse-anti-GFAP (Sigma, St. Louis MO, Cat# G3893) 1:1000; rabbit-anti-Iba1 (Wako, Richmond VA, Cat#019–19741) 1:1000; rabbit anti-Mecp2 (Cell Signaling, Danvers MA, Cat#3456S) 1:200; mouse anti-GAD67 (Chemicon, Temecula CA, Cat# MAB5406) 1:1000; mouse anti-CamKII (Abcam Cat#ab22609) 1:50; guinea pig ant-VGAT 1:1000 (Frontier, Ishikari Japan, Cat#VGAT-GP-Af1000); and rabbit anti-Vglut1 1:1000 (Frontier Cat#VgluT1-Rb-Af500). After three ten-minute washes in 1XPBS, sections were incubated overnight in secondary antibodies at 1:500 dilution as follows: goat anti-chicken Alexa 488 (Invitrogen, Carlsbad CA, Cat# A-11039); horse anti-mouse Texas Red (Vector, Burlingame CA, Cat# TI-2000); goat anti-rabbit Texas

Red (Vector Cat# TI-1000); goat anti-rabbit Alexa 488 (Invitrogen Cat# A11034); goat anti-mouse Alexa 555 (Invitrogen Cat# A-21422); goat anti-rabbit Alexa 555 (Invitrogen Cat#A-21429); and goat anti-guinea pig Alexa 555 (Invitrogen A-21435). Sections were then washed three times with 1XPBS at 10 min each wash, mounted on charged glass slides with Vectashield (Vector) mounting medium with DAPI (H-1200), coverslipped, and imaged using a confocal microscope. For synaptic puncta counts, six images were captured from each animal's CA1 using the 63X objective and maintaining the same imaging settings for each section. Images were counted for absolute puncta numbers using the Spots module of Imaris v8.0 (Bitplane, Zurich, Switzerland) and analyzed by one-way ANOVA with Tukey's post hoc analysis.

## Bone marrow/brain FACS

*Viaat*-Cre male mice were crossed with females carrying the ROSA-YFP allele. 3-week-old male pups were anesthetized with isoflurane, sacked by decapitation, and the brains and bone marrow isolated. Brains were homogenized by mincing and digestion in Dulbecco's Modified Eagle Media (DMEM) with trypsin and 0.1 mg DNAseI for 10 min at 37°C. The tissue was then run through a 22G1/2 needle 5 times and centrifuged for 10 min at 2200 rpm at 4°C. The pellet was washed with 10 mL DMEM and spun for 10 min at 1000 rpm at 4°C. The pellet was then resuspended in 1 mL Hank's Balanced Salt Solution (HBSS) and filtered through a 40 uM filter before being FACS sorted. Bone marrow was isolated from the tibia and femur and run through an 18G needle several times to dissociate the tissue and then suspended in HBSS with 10% FBS and 10mM HEPES. The cells were centrifuged five minutes at 2200 rpm at 4°C, then resuspended in HBSS and sorted. For both brain and bone marrow, 100,000 events were sorted.

## Electrophysiology

Acute fresh cortical slices were prepared from 5- to 6-week-old mice as previously described (*Lu et al., 2009*). Coronal slices (250–350 µm thick) containing somatosensory cortex were cut with a vibratome (Leica Microsystems Inc., Buffalo Grove, IL) in a chamber filled with chilled (2–5°C) cutting solution containing (in mM) 110 choline-chloride, 25 $NaHCO_3$, 25 D-glucose, 11.6 sodium ascorbate, 7 $MgSO_4$, 3.1 sodium pyruvate, 2.5 KCl, 1.25 $NaH_2PO_4$ and 0.5 $CaCl_2$. The slices were then incubated in artificial cerebrospinal fluid (ACSF, in mM) containing 119 NaCl, 26.2 $NaHCO_3$, 11 D-glucose, 3 KCl, 2 $CaCl_2$, 1 $MgSO_4$, 1.25 $NaH_2PO_4$ at the room temperature. The solutions were bubbled with 95% $O_2$ and 5% $CO_2$.

Whole-cell recording was made from pyramidal neurons in the layer II/III/V region of somatosensory cortex by using a patch-clamp amplifier (MultiClamp 700B, Molecular Devices, Union City, CA) under infrared differential interference contrast optics. Microelectrodes were made from borosilicate glass capillaries and had a resistance of 2.5–5 MΩ. Data was acquired with a digitizer (DigiData 1440A, Molecular Devices). The analysis software pClamp 10 (Molecular Devices) and Minianalysis 6.0.3 (Synaptosoft Inc., Decatur, GA) were used for data analysis. Miniature EPSCs (mEPSCs, holding at −70mV) were recorded in voltage-clamp mode in the presence of 100 mM Picrotoxin and 0.5 mM Tetrodotoxin (TTX). The intrapipette solution contained (in mM) 140 potassium gluconate, 5 KCl, 10 HEPES, 0.2 EGTA, 2 $MgCl_2$, 4 MgATP, 0.3 $Na_2$GTP and 10 $Na_2$-phosphocreatine, pH 7.2 (with KOH). Spontaneous firings were recorded in current-clamp mode (holding at −60 mV) in modified ACSF (in mM: 126 NaCl, 25 NaHCO3, 14 D-glucose, 3.5 KCl, 1 CaCl2, 0.5 MgCl2, 1 NaH2PO4). For recording mIPSCs, neurons were also held at −70 mV in voltage-clamp mode in the presence of 6-cyano-7-nitroquinoxaline-2,3-dione (CNQX, 10 mM), D-2-amino-5-phosphonopentanoic acid (AP5, 25 mM) and TTX (0.5 mM). The glass pipettes were filled with high-$Cl^-$ intrapipette solution containing (in mM) 145 KCl, 10 HEPES, 2 $MgCl_2$, 4 MgATP, 0.3 $Na_2$GTP and 10 $Na_2$-phosphocreatine, pH 7.2 (with KOH). Data were discarded when the change in the series resistance was >20% during the course of the experiment or the rest membrane potential was about −60 mV. The whole-cell recording was performed at 30 (±1)°C with the help of an automatic temperature controller (Warner Instruments, Hamden, CT). Summary data are presented as mean ± S.E.M. and analyzed by one-way ANOVA with Tukey's post hoc analysis.

## Quantitative PCR

8-week-old male mice (n = 6–7 per genotype) were deeply anesthetized with isofluorane, quickly decapitated, and their cerebella dissected and immediately frozen by liquid nitrogen. The frozen tissue was placed in 1 ml Purezol (Bio-Rad, Hercules, CA) on ice and immediately homogenized with a Polytron homogenizer. The dissolved sample was diluted to 2x using Purezol, and 1 ml of the solution was processed by Aurum Total RNA Fatty and Fibrous Tissue kit (Bio-Rad) to collect RNA. DNA within the sample was removed with on-column DNase digestion which was included in the kit. First-strand cDNA was synthesized using M-MLV reverse transcriptase (Life Technologies, Carlsbad, CA). RT-qPCR was performed using Bio-Rad CFX96 Real-Time system. The primer sets were designed to amplify target genes using UCSC genome browser and Primer 3. The same primer sets from He et al. were used to amplify *Gad1*, *Gad2*, *Pv*, *Vgat*, *Gephrin*, *Gabra1*, *Kv3.1*, and *Vglut1* (*He et al., 2014*).

To search for other candidate genes, we referred to the microarray results obtained from the cerebellum of *Mecp2*-null (*Mecp2$^{-/y}$*) and overexpressing mice (Tg3) (*Ben-Shachar et al., 2009*). The top 20 genes altered in opposite directions in the two lines were selected from the microarray results. We checked expression pattern of these genes using the Allen Brain Atlas, and we chose those genes that were clearly expressed in inhibitory neurons in the cerebellum (i.e., either Purkinje cells, stellate cells, or Golgi cells) as candidates. All RT-qPCR reactions were conducted in duplicate, and relative proportions of the cDNA were determined based on the threshold cycle (Ct). The results were averaged for each sample and normalized to the value of *Gapdh*. Relative expression levels of the target genes were determined by normalizing the fold expression level in each sample to the average of the littermate WT controls. Statistical analysis were performed on delta Ct values (i.e., [average Ct of the target gene] – [average Ct of *Gapdh*]), by one-way ANOVA followed by Tukey's post hoc test. The following primers were designed in this study:

*Gapdh*: Forward 5'- ggagattgttgccatcaacga-3', Reverse 5'- tgaagacaccagtagactccacgac-3'
*Nxph4*: Forward 5'-gtgagcacccctactttgga-3', Reverse 5'-aaggctgtttttctccacca-3'
*Gabra3*: Forward 5'-ccaccatctccaagtctgct-3', Reverse 5'-agatgatgcgggaaattttg-3'
*Kcng4*: Forward 5'-caggaggaacctcagtcagc-3', Reverse 5'-gagcccatggatatgtggac-3'
*Opn3*: Forward 5'-tggtatccctgttcggagtc-3', Reverse 5'-tcataggccagcacagtgag-3'
*Scg2*: Forward 5'-gaaatgatcagggctttgga-3', Reverse 5'-ctctctgccaagtggctttc-3'
*Ai593442*: Forward 5'-gacattgacaccgaagcaaa-3', Reverse 5'-cagctgaaggtgaggaaagg-3'
*Robo2*: Forward 5'-agcagtccactgccactctt-3', Reverse 5'-gttgtggaggtggggtattg-3'
*Rassf8*: Forward 5'-ggtcaatgaggaggaggtga-3', Reverse 5'-ctgctccttgtcctgtagcc-3'
*Cirbp*: Forward 5'-cgaagtggtggtggtaaagg-3', Reverse 5'-ccagcctggtcaactctgat-3'
*Cabp7*: Forward 5'-atggcactgactttgacacg-3', Reverse 5'-tggctctcctcctctgtcat-3'
All other primers used have been previously published (*He et al., 2014*).

## HPLC analysis of GABA

8-week-old male mice (n = 4–5 per genotype) were deeply anesthetized with isofluorane, quickly decapitated, and their striata dissected into cold 1X PBS and snap frozen. Tissue was homogenized in cold 1X PBS with a polytron and centrifuged for 5 min at 13.2 krpm at 4°C. The supernatant was collected for HPLC analysis on an amino acid analyzer (*Mayne et al., 2001*).

## Surgery and EEG recordings

The methods were modified from previous publications (*Sztainberg et al., 2015*). Adult male mice at 7–8 weeks of age were anesthetized with isoflurane and mounted in a stereotaxic frame. Under aseptic conditions, each mouse was surgically implanted with 4 silver wire (127 μm diameter, A-M Systems, Sequim, WA) recording electrodes aimed at the subdural space of the left/right frontal cortex (A2.0, LR1.8) and left/right parietal cortex (P3.8, LR1.8), respectively (*Paxinos and Franklin, 2001*). The reference electrode was positioned in the occipital region of the skull. All electrode wires were attached to a miniature connector (Harwin Connector, Portsmouth, Hampshire, UK) that was secured on the skull by dental cement. After 2 weeks of post-surgical recovery, cortical EEG activities (filtered between 0.1 Hz and 1 kHz, sampled at 2 kHz) were recorded for 1 hr per day over 4 days. Traces were analyzed by eye for abnormal electrographic signals.

## Statistical analysis

Sample sizes for all analyses were determined based on previous experience (*Chao et al., 2010*; *Chahrour et al., 2008*; *Samaco et al., 2009*, *2013*). All statistical analyses were conducted using Graphpad Prism software. Details of the analysis, including individual p values, are reported in *Supplementary file 1*.

## Acknowledgements

We thank R Samaco for mice and discussions, B Belfort for assistance with scoring data entry, P Goodell and her lab members for help with FACS, M van der Heijden for EEG analysis assistance, and V Brandt, X Meng, and L Lombardi for critical review of the manuscript. All animal behavior was carried out in the Baylor Mouse Neurobehavior Core, overseen by C Spencer, who also provided assistance with assays. HPLC analysis was done by the Baylor Analyte Center overseen by Q Sun. This work was supported by the International Rett Syndrome Foundation; NINDS 1F32NS083137-01A1 (KU); NINDS 5R01NS057819 (HYZ), the Neuroconnectivity Core of IDDRC at Baylor College of Medicine (U54 HD083092), and BCM IDDRC Grant Number 5P30HD024064-23 from the Eunice Kennedy Shriver National Institute of Child Health & Human Development. (The content is solely the responsibility of the authors and does not necessarily represent the official views of the Eunice Kennedy Shriver National Institute of Child Health & Human Development or the National Institutes of Health). WC was supported by a Startup Fund from the Gordon and Mary Cain Pediatric Neurology Research Foundation and the Jan and Dan Duncan Neurological Research Institute to Dr. Mingshan Xue. HYZ is an investigator with the Howard Hughes Medical Institute.

## Additional information

### Competing interests

HYZ: Senior editor, *eLife*. The other authors declare that no competing interests exist.

### Funding

| Funder | Grant reference number | Author |
|---|---|---|
| National Institute of Neurological Disorders and Stroke | 1F32NS083137-01A1 | Kerstin Ure |
| Intellectual and Developmental Disabilities Research Center | U54 HD083092 | Zhenyu Wu Jianrong Tang |
| Intellectual and Developmental Disabilities Research Center | 5P30HD024064-23 | Huda Y Zoghbi |
| National Institute of Neurological Disorders and Stroke | 5R01NS057819 | Huda Y Zoghbi |

The funders had no role in study design, data collection and interpretation, or the decision to submit the work for publication.

### Author contributions

KU, HL, Designed the experiments, Interpreted the data, Performed the experiments, Wrote the manuscript; WW, Performed the experiments, Conception and design, Analysis and interpretation of data, Drafting or revising the article; AI-I, Designed the experiments, Interpreted the data, Performed the experiments; ZW, Performed electrode construction, surgery, and EEG/behavioral recordings, Conception and design, Analysis and interpretation of data; L-jH, YS, Performed the experiments, Analysis and interpretation of data; WC, Assisted with EEG analysis and figure preparation, Analysis and interpretation of data, Drafting or revising the article; JT, Designed and supervised the experiments, Wrote the EEG recording methods, Analysis and interpretation of data; HYZ, Designed the experiments, Interpreted the data, Input to write the manuscript

**Author ORCIDs**
Huda Y Zoghbi, http://orcid.org/0000-0002-0700-3349

**Ethics**
Animal experimentation: All mouse care and manipulation was approved by the Baylor College of Medicine Institutional Animal Care and Use Committee (IACUC, Protocol AN-1013). Every effort was made to minimize suffering.

## Additional files

**Supplementary files**
• Supplementary file 1. Summary of statistical analyses.

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
