## [Decision Letter]

Thank you for submitting your article "Restoration of Mecp2 expression in GABAergic neurons rescues multiple disease features in a mouse model of Rett Syndrome" for consideration by *eLife*. Your article has been reviewed by three peer reviewers, and the evaluation has been overseen by a Reviewing and a Senior Editor. The following individuals involved in review of your submission have agreed to reveal their identity: Sacha Nelson (peer reviewer); Lisa Monteggia (peer reviewer).

The reviewers have discussed the reviews with one another and the Senior Editor has drafted this decision to help you prepare a revised submission.

Summary:

This manuscript reporting the consequences of selectively rescuing MeCP2 expression in GABAergic neurons is a nice complement to a previous study from the same lab reporting the phenotypic effects of losing MeCP2 in these same neurons. As in all of the studies in this series coming from the Zoghbi group, the behavioral experiments are particularly well controlled for genetic background and important effort is expended to study mosaic female mice as well as the male knockouts, since the females are more relevant to the disease condition.

A fundamental weakness of the approach is that GABAergic neurons are incredibly diverse and subserve so many functions throughout the brain that we are left with major questions about where the relevant neurons are. The authors have tended to imply that they think important phenotypes arise from cortical/hippocampal interneurons, for example by only assessing physiology in this region, but it is much more likely that all or the vast majority of the behavioural phenotypes arise from subcortical structures including the brainstem, cerebellum and striatum. Overall, with the revisions suggested below, the reviewers think that this is an important piece of work that complements the companion paper and has the potential to move the field forward.

Essential revisions:

1) In the first paragraph of the subsection “Male GABAergic rescue mice have normalized body weight and extended lifespan”: "all other cells in the body […] remain null for MeCP2" this is unlikely to be true since glycinergic neurons in the brainstem and spinal cord also expression *Viaat*. This is not a minor point because ablation of VIAAT in glycinergic neurons causes a respiratory phenotype and perinatal death (Rahman et al. 2014). *Slc32a1* is also expressed in important neuronal types that co-release GABA and other transmitters like ACh (Saunders et al. 2015) and glutamate (Root et al. 2014; Shabel et al. 2014).

Changes to reflect this more nuanced view should also be made to the Discussion. It is not clear that all of the observed effects relate to changes in inhibition as changes in the cholinergic or glutamatergic function could also occur in these mice but were not assessed.

2) As stated in the review of the companion paper, because there is ample evidence that activity of pyramidal neurons regulates inhibition and the authors have shown that activity is reduced by loss in excitatory neurons, it would be useful to test if the inhibitory rescue animals have seizures or abnormal EEGs.

3) The complementary nature of the behavioral phenotypes associated with GABAergic versus glutamatergic loss of function models is quite interesting. However, in order to be able to compare and contrast the two studies it is important to document cortical neuron activities (action potentials as well as synaptic input) as they were documented in the companion study. This manuscript shows a regulation of spontaneous miniature IPSCs and EPSCs following Mecp2 loss of function and rescue. However, the functional impact of these changes on neuronal physiology is not explored.

4) Are the changes shown in Figure 5 due to alterations in synapse numbers or functions of synaptic inputs? Earlier work from the same group (Chao et al., 2007) demonstrated an impact of MeCP2 on synapse number, therefore it is important to establish the basis of these observations.

5) It is surprising that the two studies by Goffin et al. 2014 (PMID 24777420) and He et al. 2014 (PMID 25297674) were not cited or discussed in the current manuscript, especially since the Goffin et al. paper also did selective rescue in the Pv+ population.

6) The interpretation in the Discussion is too simplistic and does not mesh very well with the companion paper. Examples are below.

Discussion, second paragraph: The fact that "This initially seems surprising, as excitatory neurons constitute the vast majority of the neuronal population" is based on a corticocentric view which does not hold brain-wide. Something like 97% of the cells in the striatum are GABAergic. In other places GABAergic neurons are key bottlenecks (e.g. Purkinje neurons).

Discussion, fourth paragraph: "When Mecp2 expression is restored in the entire population of inhibitory neurons in males, the increase in GABA signaling effectively counteracts the deficit resulting from the loss of Mecp2 in all excitatory neurons, possibly explaining the near-complete rescue of disease-related symptoms in the male" this seems excessive given e.g. the reduced lifespan, startle, anxiety, excitatory physiology and respiratory defects. Also key phenotypes were not assessed: learning and seizures stand out.

Discussion, fourth paragraph: "In the female rescue mice, however, random X-inactivation should allow roughly 50% of the excitatory and inhibitory neurons to retain functional Mecp2(Young and Zoghbi, 2004), resulting in a heterogeneity of excitatory neuron activity within the brain of female mice that may neutralize much of the benefit gained by normalizing inhibitory neuron function. This idea is supported by the work of Meng et al., who show that rescue of Mecp2 expression solely in excitatory neurons confers greater rescue in females than males.” But this ignores the fact that glutamatergic rescue in males dramatically extended lifespan to similar levels to that achieved in the *Viaat* rescue.

---

## [Author Response]

Essential revisions:

1) In the first paragraph of the subsection “Male GABAergic rescue mice have normalized body weight and extended lifespan”: "all other cells in the body […] remain null for MeCP2" this is unlikely to be true since glycinergic neurons in the brainstem and spinal cord also expression Viaat. This is not a minor point because ablation of VIAAT in glycinergic neurons causes a respiratory phenotype and perinatal death (Rahman et al. 2014). Slc32a1 is also expressed in important neuronal types that co-release GABA and other transmitters like ACh (Saunders et al. 2015) and glutamate (Root et al. 2014; Shabel et al. 2014).

Changes to reflect this more nuanced view should also be made to the Discussion. It is not clear that all of the observed effects relate to changes in inhibition as changes in the cholinergic or glutamatergic function could also occur in these mice but were not assessed.

The reviewers make a very valid point; therefore, we have softened the language in the first paragraph of the subsection “Male GABAergic rescue mice have normalized body weight and extended lifespan” and have amended the Discussion to include mention of the potential role of glycinergic and GABA/glutamate coreleasing neurons in the lateral and medial habenula, auditory cortex, and CA3. We have also included a mention of a very recent paper describing the behavioral effects of conditional deletion of *Mecp2* from cholinergic neurons. Please see the second paragraph of the Discussion for these additions.

2) As stated in the review of the companion paper, because there is ample evidence that activity of pyramidal neurons regulates inhibition and the authors have shown that activity is reduced by loss in excitatory neurons, it would be useful to test if the inhibitory rescue animals have seizures or abnormal EEGs.

We analyzed EEG traces from six rescue animals and found that the EEG traces were normal in five animals while one exhibited abnormal discharges. Because of the short time frame for revisions, we were only able to record three *Mecp2*^lox-STOP/Y^ male mice, two of which showed abnormal EEG activity similar to that reported in Chao 2010. At no point in the lifespan of any rescue animal, male or female, were behavioral seizures observed. We have added this data to Figure 1—figure supplement 2 and to the end of the subsection “Male GABAergic rescue mice have normalized body weight and extended lifespan”.

3) The complementary nature of the behavioral phenotypes associated with GABAergic versus glutamatergic loss of function models is quite interesting. However, in order to be able to compare and contrast the two studies it is important to document cortical neuron activities (action potentials as well as synaptic input) as they were documented in the companion study. This manuscript shows a regulation of spontaneous miniature IPSCs and EPSCs following Mecp2 loss of function and rescue. However, the functional impact of these changes on neuronal physiology is not explored.

We have recorded spontaneous action potentials as done in Meng et al. and find normalization of activity in the rescue animals. Please see Figure 5—figure supplement 1 and the last paragraph of the subsection “Expression of *Mecp2* in GABAergic neurons improves cortical action potential firing” for these data.

4) Are the changes shown in Figure 5 due to alterations in synapse numbers or functions of synaptic inputs? Earlier work from the same group (Chao et al., 2007) demonstrated an impact of MeCP2 on synapse number, therefore it is important to establish the basis of these observations.

We have included inhibitory and excitatory synapse number analyses in Figure 5—figure supplement 1 and in the first paragraph of the subsection “Expression of *Mecp2* in GABAergic neurons improves cortical action potential firing”. We find that rescue mice have normalized numbers of VGAT+ puncta but have no significant change in VGlut1^+^ puncta in the CA1.

5) It is surprising that the two studies by Goffin et al. 2014 (PMID 24777420) and He et al. 2014 (PMID 25297674) were not cited or discussed in the current manuscript, especially since the Goffin et al. paper also did selective rescue in the Pv+ population.

We apologize for this oversight and have included a short discussion of these papers in the second paragraph of the Discussion section.

6) The interpretation in the Discussion is too simplistic and does not mesh very well with the companion paper. Examples are below.

Discussion, second paragraph: The fact that "This initially seems surprising, as excitatory neurons constitute the vast majority of the neuronal population" is based on a corticocentric view which does not hold brain-wide. Something like 97% of the cells in the striatum are GABAergic. In other places GABAergic neurons are key bottlenecks (e.g. Purkinje neurons).

We have removed this sentence for clarity.

Discussion, fourth paragraph: "When Mecp2 expression is restored in the entire population of inhibitory neurons in males, the increase in GABA signaling effectively counteracts the deficit resulting from the loss of Mecp2 in all excitatory neurons, possibly explaining the near-complete rescue of disease-related symptoms in the male" this seems excessive given e.g. the reduced lifespan, startle, anxiety, excitatory physiology and respiratory defects. Also key phenotypes were not assessed: learning and seizures stand out.

We have softened the language of this sentence: “When *Mecp2* expression is restored in inhibitory neurons in males, the increase in GABA signaling may effectively counteract much of the deficit resulting from the loss of *Mecp2* in all excitatory neurons, possibly explaining the profound rescue of disease-related symptoms in the male mice”. As mentioned above, we provide EEG traces for the rescue mice in Figure 5—figure supplement 1.

Discussion, fourth paragraph: "In the female rescue mice, however, random X-inactivation should allow roughly 50% of the excitatory and inhibitory neurons to retain functional Mecp2(Young and Zoghbi, 2004), resulting in a heterogeneity of excitatory neuron activity within the brain of female mice that may neutralize much of the benefit gained by normalizing inhibitory neuron function. This idea is supported by the work of Meng et al., who show that rescue of Mecp2 expression solely in excitatory neurons confers greater rescue in females than males.” But this ignores the fact that glutamatergic rescue in males dramatically extended lifespan to similar levels to that achieved in the viaat rescue.

To clarify this point, we have mentioned the similar rescue of lifespan in the third paragraph of the Discussion section and have rephrased this section to say “confers greater behavioral rescue in females than males” in the fourth paragraph of the Discussion section.